# The Evolution of AI-Driven Educational Systems during the COVID-19 Pandemic

**Florin-Valeriu Pantelimon *** , **Razvan Bologa, Andrei Toma and Bogdan-Stefan Posedaru**

Department of Computer Science and Cybernetics, Bucharest University of Economic Studies,
010374 Bucharest, Romania; razvanbologa@ase.ro (R.B.); andrei.toma@ie.ase.ro (A.T.);
bogdan.posedaru@csie.ase.ro (B.-S.P.)
* Correspondence: florin.pantelimon@csie.ase.ro

**Abstract:** COVID-19 had a major impact on education, substantially stimulating the need for digital remote education. From paperback books to PDFs, from physical classes to e-conferencing, and from various traditional mechanisms of information transmission to systems that are driven by artificial intelligence and use adaptive learning approaches, all of these had to be adopted by both teachers and students. This paper analyzes the use of an adaptive learning system before and during the COVID-19 pandemic from a statistical point of view on a set of data gathered in Romania during a pilot project. The main data inputs are the number of students that enrolled for a certain course, the percentage of students that successfully completed it, and information about their age group, location and main area of interest. Our study finds that the use of artificial intelligence has increased during the COVID-19 pandemic and, by analyzing the data obtained during the study, we managed to prove that artificial-intelligence-driven tools and systems have gained traction among all the parties involved in the educational process.

**Keywords:** COVID-19; artificial intelligence; education; adaptive learning

## 1. Introduction

The COVID-19 pandemic changed the way educational systems work and rewrote the rules on how students and teachers interact and how education takes place. Online education is becoming common and is a way of preparing the next generation for online work. Both teachers and students had to adapt and discover new ways of interacting, as well as utilizing existing useful, but previously overlooked, tools.

Artificial intelligence represents one of the most important ways to improve education and to adapt it to the needs of Industry 4.0.

The objective of this paper is to prove that the use of artificial-intelligence-driven education systems has grown during the COVID-19 pandemic, when most of the schools and universities around the world had to switch from physical schooling to remote schooling.

In order to prove that the use of artificial-intelligence-driven educational tools has grown, we performed a literature review, and we also gathered data from a pilot project that took place in Romania.

We consider that our research conclusions are important because most education will be digitalized over the coming years, and our research may be useful for those who will design online educational platforms. The current article opens new perspectives on what the future of education might look like. Therefore, this paper serves as proof that there is likely to be an increase in the demand for newer, artificial-intelligence-driven systems.

## 2. Literature Review

The idea of using artificial intelligence in education is not new. According to [1], tutoring systems that aimed to automatically adapt to the needs of students existed as

early as 1991. The author identified 47 articles that discuss the subject of education and AI between 1991 and 2016.

Before COVID-19, artificial intelligence had limited use in education [2]. AI algorithms were used in either experimental projects or for detecting plagiarism. Bibliometric analysis indicates that artificial intelligence has been constantly growing since the 1970s and has reached a level of maturity [3].

COVID-19 is boosting the use of online education, which represents a chance for introducing advanced AI technologies such as chatbots and text-to-speech [4] in schools. AI in education is currently a subject of interest in the scientific community. According to [5], adaptive intelligent tutoring systems are becoming more popular and have increased efficiency.

Intelligent tutoring systems are attractive because they can offer a certain degree of personalized learning. Because it is difficult to change the characteristics of students, it is easier to create learning environments that adapt to individual needs [6].

Users can also share information and populate knowledge bases [7]. By bringing together input from multiple participants, lessons can be improved, and the knowledge bases behind them become more useful to other students.

Adaptive tutoring systems also facilitate the use of web-based content and gaming content, which improve the attractiveness of the learning process [8]. AI can also determine the factors that influence the performance of students and provide descriptive information in order to improve the quality of education [9].

The classroom environment is also changing as a result of the introduction of digital education. Advanced technologies using more complicated scientific educational experiments are also supported by machine learning [10]. Teachers can integrate more advanced cross-disciplinary elements in their lessons.

The large-scale introduction of online learning has stimulated the use of artificial intelligence and makes it possible to create modern content that is attractive to students [11]. Intelligent tutoring systems have a positive impact on education.

As per [12], online learning is the form of education that takes place over the internet. The two most used forms nowadays are video conferencing and online platforms that support different types of content (text, audio, video) and are accessed both synchronously and asynchronously.

The main benefit of switching to an online approach to education is that students have more control over their activity, as they are responsible for gathering all the information, analyzing, and learning, while teachers have the responsibility of offering guidance, coaching, and stimulating students' curiosity and will to learn.

Online learning and artificial intelligence are linked because it is the digitalization of education that stimulated the introduction of machine learning in the educational process [11]. AI makes it possible to transform online learning and make it more attractive.

A report [13] analyzed the current trends around the world regarding online college education. The report states that the higher education offer has been on a descending path for the past decade. However, growth in online enrollment for higher education forms and institutions was observed for the same period. Therefore, competition continues to rise as universities continue to launch online programs. Over the next three years, more than 70% of colleges and universities that were analyzed in the report plan to launch between one and four new online programs.

The key findings of the aforementioned report demonstrate that the entire educational system needs to evolve to satisfy students' needs in the context of continuously evolving technologies, shorter attention spans, and less time to be spent on activities that do not provide any value, such as commuting.

As per [13], atotal of 59% of the respondents state that the subject is the most important factor they take into consideration when deciding, 25% state that they value the format (in-class vs. online) and the remaining 16% consider that the university is the most important factor when deciding to enroll.

Moreover, the same report [13] states that when the program the students want is not available at a certain college or university, 52% will look at the same program at different universities, while only 29% would enroll in an on-campus program at the chosen university. The remaining 19% would take into consideration enrolling in a different online program at the same university.

Findings of report [13] prove that schools and universities risk losing potential students if they do not provide online courses as an alternative to their on-campus programs. When students do not find an online program to fit their needs, they will look elsewhere instead of choosing another program at the same university.

Moreover, the authors demonstrate that speed is a very important factor when choosing an online program. A total of 21% of online college students submit their first application in less than two weeks from beginning the search, and 26% between two and four weeks. Almost half of the respondents stated that they submit their first application in the first month, and they expect the same pace when receiving the results. Furthermore, the same report shows that learning on the go is another important aspect valued by online college students, as they want to progress through their courses no matter their location or their devices (either phone or tablet), which demonstrates that technology impacts how students would like to interact with their schools, and that institutions should adapt their processes for a shorter time of completion for different tasks.

A survey conducted between March and April 2020 by the International Association of Universities [14] addressed a series of questions to 424 universities and other Higher Education Institutions (HEIs) around the world, based in 109 countries and two Special Administrative Regions of China (Hong Kong and Macao). The questions asked by the authors were mainly related to the impact of the measures against COVID-19 on the Education Act.

When questioned about how COVID-19 has affected teaching and learning, 2% of the respondents said that it was not affected; 7% said that the teaching had been totally canceled; 24% said that it had been suspended, but that the university was working on developing solutions to continue (through digital or self-study means); and the remaining 67% said that classroom teaching had been replaced by distance teaching and learning.

The aforementioned report states that many of the respondents see the transition towards remote teaching as an opportunity to propose more flexible learning opportunities, explore blended or hybrid learning, and mix synchronous learning with asynchronous learning. Additionally, some respondents expect to see an increase in innovation in the field of teaching pedagogies as well as in delivery modalities of teaching and learning [14].

Work in [15] analyzes the impact of COVID-19 on education from different perspectives, such as schools, families, assessments, and graduates, and also offers different solutions to the identified issues. Thus, from the family's standpoint, there is increased pressure on the information that the students are obtaining from home and their interactions with the family. Moreover, students tend to forget what they are supposed to be doing during their study-at-home time.

When analyzing assessments, it appears that many exams were postponed or canceled. The paper presents the case of the United Kingdom, for example, where exams for the main public qualifications were canceled. However, the paper provides evidence that, in some cases, students may benefit from those interruptions over the long term.

Paper [16] analyzes the applications of artificial intelligence during the COVID-19 pandemic. The article provides evidence that AI has played an important role in monitoring and tracking the spread of the virus, having different applications such as early detection and diagnosis of infection, monitoring treatment, projection of cases and mortality, development of drugs and vaccines, and reducing the workload of healthcare workers.

The main applications of AI in the COVID-19 pandemic are represented by early detection and diagnosis of infection by analyzing symptoms, or the output from medical imaging technologies such as CT (computed tomography). Another application is represented by monitoring treatment and providing daily updates regarding patients' health

conditions. Contact tracing of individuals represents the third application of AI in the COVID-19 pandemic, and it can be performed by analyzing virus clusters, transmission patterns, and case localization.

Moreover, with the help of AI, the projection of cases and mortality can be completed more easily, since AI gathers data about the spread and risk of infection from various sources and predicts the number of possible cases and deaths in different locations.

Paper [17] analyzes the effect of switching to interactive platforms for holding weekly educational conferences of the emergency medicine residency programs in the USA. The paper states that the association launched an interactive platform for live streaming video presentations together with a backchannel for discussions. The measures taken consisted of switching to the Zoom teleconferencing application for streaming, and to YouTube Live and Slack for discussions with educators and speakers.

The conclusions of the aforementioned paper are that, even though at first it is challenging to switch to a new approach for day-to-day activities, their event highlighted a successful, scalable, and engaging way to host a live conference by supporting real-time interactions between learners and speakers. The resident feedback states that 84% of residents felt that the newly created platform was the same or better in quality.

Paper [18] analyzes the way COVID-19 influences the switch towards a smart multi-modal education, and presents the importance of using both artificial intelligence and the Internet of Things in order to achieve smart, adaptive, resource-enriched systems that the students can make use of. Therefore, the main benefits of switching to a smart form of education are the following: saving time by using quick communication channels, increasing productivity by sharing information in different formats coming from different sources, high flexibility in the interaction between the students and teachers, improving learning performance by providing various forms of presenting data, and interactive reliability through easy communication between different parts.

Data collected during the pandemic period by other authors [19] indicates that students perceive intelligent tutoring systems as useful and easy to use. Political support is also important in the process of introducing ITS on a large scale.

The profile of teachers is also important when adopting intelligent tutoring systems [20]. Science teachers tend to be more open, as they can improve the student experience by designing attractive lessons.

However, smarter forms of education come with some challenges as well: technical challenges in hardware, software, and internet availability, the digital content challenge, and the Human–Computer Interaction (HCI) challenge.

## 3. Materials and Methods

In order to analyze the relationship between people's willingness to join and use AI-driven educational platforms and the vaccination rate, we gathered data from a wide range of sources that provide insights into the pandemic situation from different perspectives.

The first dataset was obtained using an experimental pilot project that represents an AI-driven educational system aimed at Romanian students, through which they study Robotics and other related subjects (Physics, Electronics, Programming). The study covered 1600 teachers and their students aged 8 to 16 over a period of two years.

The second dataset provides information about the vaccination process against COVID-19 in Romania.

NextLab represents an educational platform that was initially created for the study of Robotics with an AI-driven tutoring system. The use was extended to other STEM subjects, such as Physics, Electronics, and Informatics. The main benefits of this platform are represented by the ability of the teacher to create learning paths that can be followed asynchronously by students, and also creating knowledge bases that can be consulted in a highly interactive way (with the personal assistant having implemented text-to-speech functionality).

### 3.1. Research Hypothesis

The main objective of this study was to analyze the impact of COVID-19 on the adoption of AI-based intelligent tutoring systems (ITS) in Romania. The results can be extrapolated to other similar countries. The authors assume that the more developed areas tend to adopt intelligent tutoring systems faster. From this point of view, this article will correlate the adoption rates of ITS with vaccination rates, which are known to be higher in more developed areas.

The research hypothesis of this paper is that people in less developed rural areas reject modern educational technologies in the same way they rejected COVID vaccines.

In order to analyze this relationship, we gathered data generated by an AI-based e-learning platform during 2019 and 2020, before and during the COVID-19 pandemic, and data regarding the vaccination process in Romania, which represents a sensitive subject in the social environment.

Vaccination rates are used as an indicator of development of each geographical area because they also indicate the citizens' level of confidence in science. We consider this indicator to be a better choice than conventional plain vanilla indicators such as GDP per capita, or the human development index, which do not measure the level of confidence in science that people have. It is to be noted that in Romania, like all other European Union countries, there are sufficient COVID-19 vaccine doses for all eligible citizens.

The pandemic has had a profound effect on the way various social systems operate, affecting multiple facets of society. The social scene has been filled with discussions about social responsibility, health and safety, and education continuity in times of a pandemic state.

As a premise, which was analyzed in various papers and studies, the use of online learning has increased, frequently becoming the standard method of interaction between teachers and students. Therefore, we aimed to demonstrate the relationship between people's appetite for discovering new learning technologies and the vaccination rate.

### 3.2. Statistical Data

In order to test the research hypothesis, we analyzed the number of students and teachers enrolled in the pilot platform, taking into account data regarding geographical distribution. All records indicated the cities and counties of those students and teachers.

Figure 1 shows how the coverage of the application has evolved from 2019, before the COVID-19 pandemic, until 2020, during the pandemic. In 2019, 35 out of 42 counties had at least one city enrolled in the platform, while in 2020, the application gained full coverage over Romania in terms of counties. Moreover, in 2019, 87 different cities were enrolled, and this number grew in 2020 to 258, an increase of almost 300% compared to 2019.

Figure 2 shows the evolution of the number of teachers and students during the pandemic. As can be seen, in 2019 there were 230 teachers and 2216 students enrolled, and in 2020 the number of teachers grew almost twelve times to 2647, while the number of students grew almost nine times to 19,796.

Table 1 shows the main data input that was used for our statistical measurements, and it contains the evolution of both teachers and students enrolled in the application, calculated as the difference between 2020 and 2019 values and the vaccination rate of each specific county.

After identifying the growth in the number of teachers and students for each specific county and the rate of vaccination, we calculated the correlation for each pair of these values using Microsoft Excel's correlation functionality and obtained the results in Table 2 in the following section.

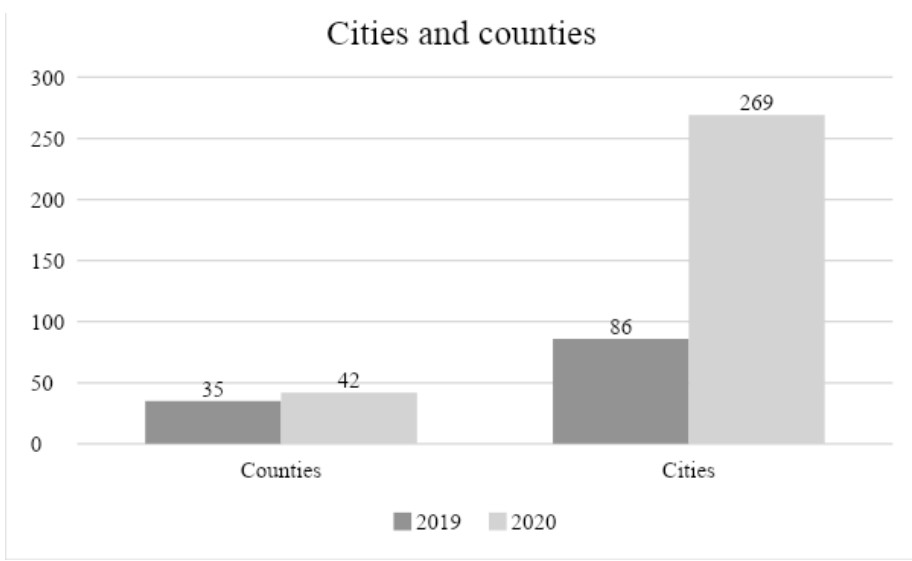

**Figure 1.** Number of cities and counties that had at least one student or teacher enrolled in the platform. Source of data: [21].

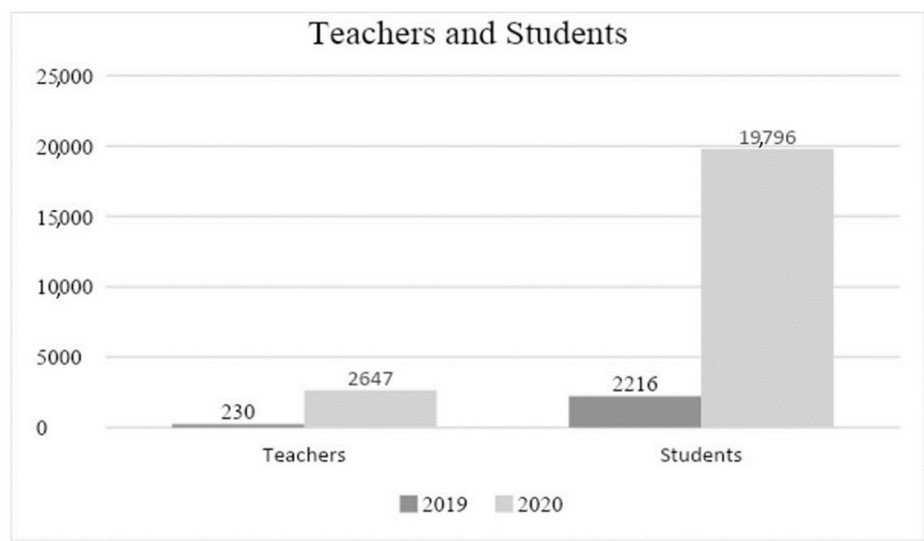

**Figure 2.** Number of teachers and students enrolled in the application in 2019 and 2020. Source of data: [21].

**Table 1.** Growth of enrolled teachers and students and vaccination rates in Romanian counties. Source of data: [21].

| County | Teachers (2020–2019) | Students (2020–2019) | Vaccination Rate (%) |
|---|---|---|---|
| Alba | 49 | 395 | 35 |
| Arad | 36 | 262 | 29.55 |
| Arges | 117 | 633 | 29.47 |
| Bacau | 24 | 194 | 22.52 |
| Bihor | 111 | 773 | 29.87 |
| Bistrita Nasaud | 17 | 99 | 31.28 |
| Botosani | 38 | 366 | 22.34 |
| Braila | 24 | 352 | 25.33 |
| Brasov | 51 | 571 | 37.96 |
| Bucuresti | 211 | 2367 | 52.77 |
| Buzau | 49 | 214 | 25.51 |

**Table 1.** *Cont.*

| County | Teachers (2020–2019) | Students (2020–2019) | Vaccination Rate (%) |
|---|---|---|---|
| Calarasi | 38 | 286 | 27.41 |
| Caras Severin | 9 | 64 | 29.24 |
| Cluj | 121 | 925 | 47.85 |
| Constanta | 41 | 288 | 39.82 |
| Covasna | 4 | 22 | 21.88 |
| Dambovita | 52 | 248 | 28.65 |
| Dolj | 35 | 266 | 33.23 |
| Galati | 32 | 318 | 30.68 |
| Giurgiu | 8 | 49 | 21.07 |
| Gorj | 54 | 144 | 27.87 |
| Harghita | 25 | 101 | 26.05 |
| Hunedoara | 72 | 184 | 32.19 |
| Ialomita | 14 | 67 | 28.66 |
| Iasi | 53 | 442 | 31.82 |
| Ilfov | 143 | 1123 | 36.68 |
| Maramures | 113 | 922 | 30.45 |
| Mehedinti | 17 | 201 | 24.43 |
| Mures | 44 | 206 | 34.16 |
| Neamt | 66 | 616 | 26.15 |
| Olt | 40 | 259 | 26.45 |
| Prahova | 161 | 1540 | 33.44 |
| Salaj | 34 | 194 | 34.16 |
| Satu Mare | 38 | 187 | 33.42 |
| Sibiu | 48 | 358 | 40.96 |
| Suceava | 92 | 600 | 20.6 |
| Teleorman | 25 | 181 | 25.76 |
| Timis | 78 | 287 | 38.95 |
| Tulcea | 17 | 150 | 32.3 |
| Valcea | 68 | 479 | 32.44 |
| Vaslui | 74 | 347 | 25.13 |
| Vrancea | 74 | 302 | 23.14 |

**Table 2.** Correlation coefficients between teachers and students growth and the vaccination rate. Source: [21].

| | Teachers (2020–2019) | Students (2020–2019) | Vaccination Rate |
|---|---|---|---|
| Teachers (2020–2019) | 1 | | |
| Students (2020–2019) | 0.92 | 1 | |
| Vaccination rate | 0.53 | 0.57 | 1 |

## 4. Results

Our data come from user registration and activity on an AI-driven learning platform. The data contain registration dates and locations from the entire territory of Romania, segmented by the location of the registered users over two years. Despite the small (from a temporal standpoint) number of data points, the increase is remarkable enough to be relevant.

The data show an increase in the total number of participants from the year 2019 to 2020 of 917%.

The increase in the total number of teachers enrolled in the platform went from 230 in 2019 to 2647 in 2020, which represents a growth of 1150%. For students, this increase was from 2216 in 2019 to 19,796 in 2020, which represents a growth of 893%.

From the above data, there was a significant increase in both the number of students and teachers enrolled in the platform, independent of their geographical location and also independent of the originating environment, either rural or urban.

Moreover, in terms of the correlation between people's preference for adopting AI-driven educational platforms and their willingness to become vaccinated against COVID-19, we have obtained the correlation coefficients presented in Table 2. We found R = 0.53 for the relationship between the growth in the number of teachers and the vaccination rate for a specific county, and R = 0.57 which represents a moderate positive relationship between these values. This means that teachers and students who are willing to adopt AI-driven educational platforms are slightly more likely to be vaccinated against COVID-19.

This indicates that there is a moderate correlation between the adoption of ITS and the general level of development of each geographical area (indicated by the vaccination rate).

This result seemed to indicate that less-developed areas with less science-oriented inhabitants were more open to modern educational technologies than they were to COVID-19 vaccines. The fact that the correlation coefficient is around 0.5 gives hope that less-science-educated citizens will not reject modern AI-based educational technologies in the same way they rejected COVID-19 vaccines. Figure 3 expresses the correlation between the growth in the numbers of students and teachers (absolute values 2020–2019) and the vaccination rate (%) for every county in Romania.

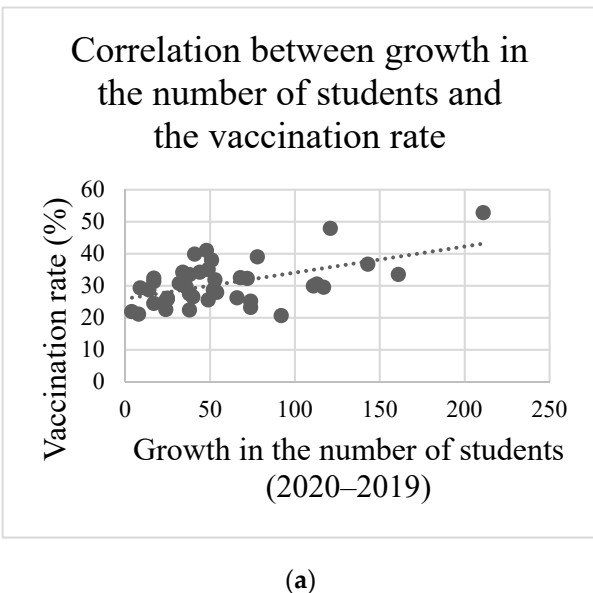

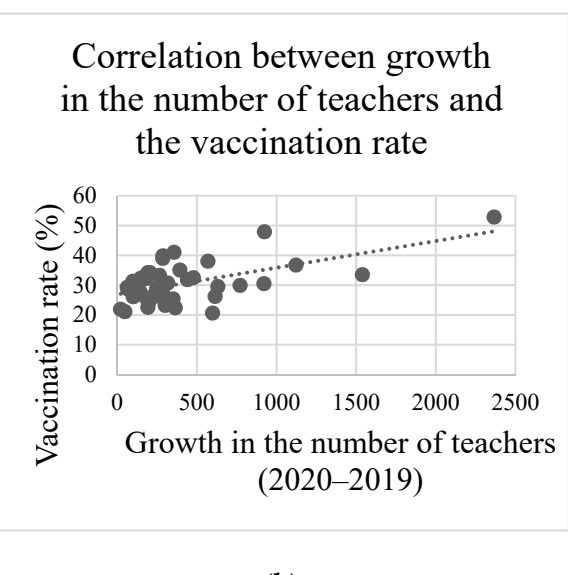

(**a**)　　　　　　　　　　　　　　　　　　　(**b**)

**Figure 3.** (**a**) Correlation between growth in the number of teachers and the vaccination rate; (**b**) Correlation between growth in the number of students and the vaccination rate.

When comparing our results with the ones obtained by other studies, we can see that, for example, ref. [19] concluded that students' preference for adopting intelligent tutoring systems (ITS) is positively influenced by politics and the perceived value of these systems.

Moreover, ref. [22] analyzes the influence of poverty and other socio-economic factors on the vaccination rate, with the study's results showing that negative socio-economic indicators, such as a lack of insurance, firearm fatalities, and housing problems, have a significantly negative influence on the vaccination rate.

Therefore, we can confidently conclude that people's willingness to become vaccinated against COVID-19 serves as an indicator of the population's welfare.

## 5. Conclusions and Future Work

In order to confirm our research hypothesis, we analyzed multiple sets of data regarding the use of both AI-driven and non-AI-driven educational systems and the increased adoption of these types of platforms.

According to the data presented in this paper, we can observe that the use of AI-driven platforms increased from 2019 to 2020, independent of both geographical and

demographical data. We can see this increase for both teachers and students, which demonstrates a common effort made by both parties towards the evolution of traditional educational systems.

Based on the increased use and adoption of AI-driven educational platforms, we can conclude that the hypothesis was confirmed: we can observe an increased interest in AI-driven educational solutions, with a higher use of such platforms even in less-developed geographical areas.

We consider the findings of this statistical experiment to be useful for those who design educational policies. Due to technological progress, the labor market will change in the near future. This implies that there is a strong need to profoundly reform current educational systems by introducing modern technologies that can support the development of skills relevant for the jobs of the future.

The good news provided by our research is that less-developed areas are still open to adopting educational technologies in the process of educating future generations.

In terms of future perspectives, we are constantly monitoring the vaccination rate growth, as well as the number of enrolled people, either students or teachers, and consider extending the research to a wider geographical area, such as the European Union. However, the main challenge, in this case, is the availability of both vaccination progress and AI-driven educational systems' adoption rate.

**Author Contributions:** Conceptualization: R.B.; methodology, R.B.; software, A.T.; validation, A.T.; investigation, F.-V.P.; data curation, A.T.; writing—original draft preparation, F.-V.P.; B.-S.P.; writing—review and editing, R.B.; A.T.; visualization, F.-V.P.; supervision, R.B.. All authors have read and agreed to the published version of the manuscript.

**Funding:** This research received no external funding.

**Institutional Review Board Statement:** Not applicable.

**Informed Consent Statement:** Not applicable.

**Data Availability Statement:** The data is available and can be quoted at: Toma, Andrei (2021), "Nextlab.tech contest participant data 2019–2020", Mendeley Data, V1, doi:10.17632/trb42h4ktc.1.

**Conflicts of Interest:** The authors declare no conflict of interest.

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
