# Peer review of "The Evolution of AI-Driven Educational Systems during the COVID-19 Pandemic"

_sustainability, doi:10.3390/su132313501_

Round 1
Reviewer 1 Report
I consider that the paper addresses an interesting and also current topic, AI driven educational solutions before and during the pandemic.
However, I would have wished for a more detailed development of the studied subject, therefore please allow me to make some friendly recommendations for the authors in order to improve their article so as to be published in this journal:
-The list of bibliographic references on which it is based is extremely short - I recommend extending it (and implicitly the section "2.Literature review") with other relevant studies in the field studied, in order to support the research conducted by authors
-Although I have nothing against the charts presented the Introduction, I would recommend the authors to be clearer in describing the source / experiment on which figures 1-4 are based, because apparently it seems to be based on the "report [2]" paragraph 50-57 (and also appear to be copy-pasted in Turnitin check). If they are really part of the authors' own experiment, maybe these figures have their place in another section.
- In the Materials and Methods section (wrongly numbered with 2. instead of 3., which denotes a little inattention on the part of the authors) I would recommend a more detailed description (more than 2 short sentences) of the first and second dataset and how they were collected.
- There is another subsection called 2.2 Statistical Data, but apparently the statistical experiment is missing (or the authors missed to present it in this manuscript), apart from some functions like SUM / COUNT presented in figures 5 and 6.
- I would recommend the improvement of the presented figures, at least as a design aspect (because they seem to be some very simple figures generated in Excel). Futhermore, other more complex statistical computations can be introduced in the manuscript based on the two datasets
- I would also recommend the development and extension of both the Materials and Methods section and the Results section (this one having just 12 lines, 211-223) since I believe they are insufficient for the readers to realize the magnitude of the experiment, the work done by the authors, and the impact of the results obtained.
- in the Conclusions section I would have liked to see the future plans of the authors related to the extension of the present study.
The article seems original, at a Turnitin antiplagiarism check (see attached file) there is a 18% similarity. However, there are certain parts that appear colored by Turnitin check (especially the figures 1-4), and therefore unfortunately taken without citation, to which I would recommend the authors to rephrase them using their own words or to put them in quotation marks with the cited reference next to them.
The article appears at the beginning as interesting, but apparently somehow too brief thus affecting a better understanding of the quality and contribution of the authors in the researched field. I would recommend the authors to consider the observations made in order to improve their manuscript for the publication in this journal.

Author Response
Please find attached the answers.

Reviewer 2 Report
I suggest including the sentence in the Abstract: "The purpose of the study was ...". It is true that the authors wrote "This paper analyzes the usage ...", but in my opinion such an approach is not sufficient. On the other hand, the authors formulated the aim of the work in several places in the article. On lines 29-30 there is the sentence "... the main goal of this paper is to prove that the usage of Artificial Intelligence ...". In line 171 the authors wrote “The main objective of this study is to analyze the impact…”. In line 177 the authors wrote “The general objective of our paper is to analyze the impact…”. I think that the purpose of the work should be written once and in the right place. Stating a study goal several times creates unnecessary confusion.
I do not understand the structure of the article and the layout of the individual chapters at all. After the Introduction chapter, the Authors formulate the chapter Literature review. After all, the literature review is made in the Introduction in order to formulate a research problem on this basis, which in turn is the basis for presenting the research goal. In the Introduction, the Authors cite some research results that should not be included here at all. The results of research by other research teams could be used in the discussion of the results of own research.
Section 2.1 is entitled Research Hypothesis. In this subsection, however, I could not find the exact formulation of the research hypothesis anywhere. The authors mentioned the confirmation or denial of the research hypothesis in Conclusions, but it is not known exactly what hypothesis they wanted to confirm.
Presentation of the research results on 1/3 of the page is unacceptably short and too laconic. When presenting the research results, the authors refer to the data presented in Materials and Methods. I don't understand it at all.
In the section Material and Methods please indicate what kind of methods and model research were used by Authors. In scientific research we can use two types of models and the resulting research methods and techniques. Please refer to articles that describe and use different models and methods and please reinforce and support this section of the article with the quotes mentioned: Significance and directions of energy development in African countries. Energies, 2021, 14(15), 4479 and next paper: New technologies and innovative solutions in the development strategies of energy enterprises. HighTech and Innovation Journal, 2020, 1(2), 39-58. Creswell, J.W. Educational research: Planning, conducting, and evaluating quantitative and qualitative research, 3rd ed.; Upper Saddle River: New Jersey, USA, 2008. Thanks to this, readers will have the opportunity to familiarize themselves with the research methods.
The article includes only 7 source materials in its references. This is definitely not enough for a research paper. After all, in the world literature you can find a lot of articles on contemporary problems of education during a pandemic, and it is worth including these articles in a properly conducted literature review. For example, you can refer to the articles: "Online learning: A panacea in the time of COVID-19 crisis", "The topic of the ideal dairy farm can inspire how to assess knowledge about dairy production processes: A case study with students and their contributions", "A literature review on impact of COVID-19 pandemic on teaching and learning".
Generally, the article is written very chaotically and without taking into account the rules of writing scientific articles.
I suggest rewriting the article in an orderly manner, with a neatly conducted literature review and taking into account the structure adopted when writing a scientific article.
Author Response
Please find attached the answers.

Reviewer 3 Report
- Where is the data in Figures 1-4 from? What is the relationship between these figures and the rest of the paper?
- The general objective of this manuscript is to analyze the impact the pandemic had on the educational process and educational methodologies. To achieve this goal, large amount of data from various sources should be provided. However, the only data source in this paper is an AI driven educational system in Romania, and the volume of the data is not enough to support the point presented in the manuscript.
- Furthermore, the increase in the number enrolled in the platform and the number of countries does not necessarily means that the increasing interest and impact AI driven technologies have had on education.
Author Response
Please find attached the answers.

Round 2
Reviewer 1 Report
The authors addressed point by point all the modifications / suggestions I made, in my opinion thus adding a plus value to the manuscript. Of course, there is always room for improvement, but I appreciated the benevolence, openness of the authors, and their desire to improve their research and manuscript for publication.
The degree of similarity dropped to 14% in Turnitin from initial 18% in the first version of the manuscript, which I think is fine.
There are a number of other aspects of the manuscript formatting (for example table 1 which is broken on two pages) but these can be solved together with the team of editors.

Author Response
Dear Mr./Mrs.,
We would like to express our gratitude for your effort in reviewing our paper.
In order to lower the degree of similarity, we rephrased the first four figures, as they were reproduced by the article cited.
Moreover, we have updated the article based on the requirements of another reviewer as follows:
- We have updated the section 3.1. Research Hypothesis and introduced the following phrase The research hypothesis of this paper is that people in less developed rural areas reject modern educational technologies in the same way they rejected COVID vaccines. In order to emphasize the research hypothesis of our paper.
- We have updated the respective paragraph (lines 261-264) with the correct data.
- We did some reformulations in the Results chapter in order to incorporate the correlation table.
- We updated the two charts from figure 7 and included axis names.
- We have added information about other papers that analyzed the same subject from other points of view at the end of the Results section.
- We have rewritten the entire reference list in accordance with the requirements.
Best regards,
Florin Pantelimon
Reviewer 2 Report
In subsection 3.1. Research Hypothesis I did not find a precisely formulated research hypothesis. If the subchapter includes the word Hypothesis in the title, I expect to find the sentence: Hypothesis H1 is as follows ... Please complete this element of the article to refer to the hypothesis in the research part and confirm or deny it. In Chapter 5 (Conclusions and future work) the Authors wrote "... the hypothesis was confirmed ..." (line: 326), but I still do not know what hypothesis the Authors formulated and related to what they confirmed.
On lines 262-265 the authors wrote: "Figure 6 show the evolution of teachers and students numbers during the pandemic. As it can be seen, in 2019 there were 231 teachers and 2249 students enrolled and in 2020, the teachers number has grown almost ten times to 2249, while the students number has grown 8.23 times to 18525 ”. In Figure 6, however, I could not find the data given in the sentence. Please clarify and correct the data.
It seems to me that the analysis of the correlation coefficients presented in Table 2 should be presented in the Results chapter, not in the Materials and Methods chapter.
In Figure 7 (a and b) the abscissa (x) axes and the ordinate axes (y) require a description of what data are presented on the axes.
In a classic scientific article, the research results are developed by a discussion in which the results of own research are compared with the results, as well as the opinions presented in the literature on the subject. In the Results chapter, the authors never once cited other publications to discuss the analyzed research problem. This part of the article needs to be supplemented.
In the publications listed in References, the names and then the first names (first letters of the given names) of the authors of the articles are given. In this article, these rules have not been respected, therefore the materials in References need to be corrected. In addition, DOI numbers must be added to the articles in References (if possible).
Author Response
Please see the attached response.

Reviewer 3 Report
The revised manuscript has been improve according to the suggestions. I have no more comments.
Author Response

(The authors gave the same response as above.)

Round 3
Reviewer 2 Report
Thank you for your responses to the comments in the review. Why is there the same number of cited publications in the last version of the article as in the first version of the article (7 citations)? In the second version of the article there were many more citations (21 citations)!
Author Response
Dear reviewer,
I downloaded the manuscript and indeed it seems that it's an earlier version of it.
Please check the updated one.
Thank you very much for your time.
Kind regards,
Florin Pantelimon